

**High-resolution quantification of atmospheric CO₂ mixing ratios in the Greater Toronto Area,**
**Canada**
Stephanie C. Pugliese[1], Jennifer G. Murphy[1*], Felix R. Vogel[2,4], Michael D. Moran[3], Junhua Zhang[3],
Qiong Zheng[3], Craig A. Stroud[3], Shuzhan Ren[3], Douglas Worthy[4], Gregoire Broquet[2]
[1] University of Toronto, Department of Chemistry, 80 St. George St, Toronto, ON, Canada M5S 3H6
[2] Laboratoire des Sciences du Climat et de L'Environnement, CEA-CNRS-UVSQ, Université de Paris-
Saclay, France
[3] Environment Canada, Air Quality Research Division, 4905 Dufferin St. Toronto, ON, Canada M3H
5T4
[4] Environment Canada, Climate Research Division, 4905 Dufferin St. Toronto, ON, Canada M3H 5T4
*Correspondence author. Email address: jmurphy@chem.utoronto.ca (J.G. Murphy)



**Abstract**
Many stakeholders are seeking methods to reduce carbon dioxide ($CO_2$) emissions in urban areas,
however reliable, high-resolution inventories are required to guide these efforts. We present the
development of a high-resolution $CO_2$ inventory available for the Greater Toronto Area and
surrounding region in southern Ontario, Canada (area of $\sim 2.8 \times 10^5$ km², 26 % of the province of
Ontario). The new SOCE (Southern Ontario $CO_2$ Emissions) inventory is available at the 2.5 x 2.5 km
spatial and hourly temporal resolution and characterizes emissions from seven sectors: Area,
Residential natural gas combustion, Commercial natural gas combustion, Point, Marine, On-road and
Off-road. To assess the accuracy of the SOCE inventory, we developed an observation-model
framework using the GEM-MACH chemistry-transport model run on a high-resolution grid with 2.5
km grid spacing coupled to the Fossil Fuel Data Assimilation System (FFDAS) v2 inventories for
anthropogenic $CO_2$ emissions and the European Center for Medium-Range Weather Forecasts
(ECMWF) land carbon model C-TESSEL for biogenic fluxes. A run using FFDAS v2 for the southern
Ontario region was compared to a run in which its emissions were replaced by the SOCE inventory.
Simulated $CO_2$ mixing ratios were compared against in situ measurements made at four sites in
southern Ontario, Downsview, Hanlan's Point, Egbert and Turkey Point, in three winter months,
January-March, 2016. Model simulations had better agreement with measurements when using the
SOCE inventory emissions versus other inventories, quantified using a variety of statistics such as
correlation coefficient, root mean square error and mean bias. Furthermore, when run with the SOCE
inventory, the model had improved ability to capture the typical diurnal pattern of $CO_2$ mixing ratios,
particularly at the Downsview, Hanlan's Point and Egbert sites. In addition to improved model-
measurement agreement, the SOCE inventory offers a sectoral breakdown of emissions, allowing
estimation of average time-of-day and day-of-week contributions of different sectors. Our results
show that at night, emissions from Residential and Commercial natural gas combustion and other



Area sources can contribute > 80 % of the $CO_2$ enhancement while during the day emissions from the
On-road sector dominate, accounting for >70 % of the enhancement.

**1.0 Introduction**

Urban areas are sites of dense population and the intensity of human activities (such as
transportation, industry and residential and commercial development) makes them hot-spots for
anthropogenic carbon dioxide ($CO_2$) emissions. While occupying only 3 % of the total land area, urban
areas are locations of residence for 54 % of the global population and are the source of 53 – 87 % of
anthropogenic $CO_2$ emissions globally (IPCC-WG3, 2014; WHO, 2015). When considering Canada
alone, the urban population accounts for an even larger fraction of the total (81 % in 2011) (Statistics
Canada, 2011) while urban areas cover only 0.25 % of the land area (Statistics Canada, 2009).
Recognizing their influence on the global carbon budget, many urban areas are seeking methods to
reduce their anthropogenic $CO_2$ emissions. The Greater Toronto Area (GTA) in southeastern Canada,
for example, has committed to the *Change is in the Air* initiative as well as being a part of the *C40 Cities*
*Climate Leadership Group,* both of which call to reduce $CO_2$ emissions 30 % below 1990 levels by 2020
(C40 Cities, 2016; Framework for Public Review and Engagement, 2007). However, in order to
effectively guide anthropogenic $CO_2$ mitigation strategies, reliable inventories are needed,
particularly at high spatial and temporal resolution, to gain a better understanding of the carbon
cycle (Gurney et al., 2009; Patarasuk et al., 2016). To our knowledge, the only spatially disaggregated
$CO_2$ inventories available for use in the GTA are the EDGAR v.4.2 (Emission Database for Global
Atmospheric Research) $CO_2$ inventory (available at annual, $0.1^o$ x $0.1^o$ resolution) (EDGAR, 2010) and
the FFDAS v2 (Fossil Fuel Data Assimilation System) $CO_2$ inventory (available at hourly, $0.1^o$ x $0.1^o$
resolution) (FFDAS, 2010), both which are limited in their spatial and/or temporal resolution and
therefore are not well-suited for the quantification and understanding of $CO_2$ emissions at the urban





scale. The Canadian national $CO_2$ inventory, on the other hand, is only available at the provincial level
(Environment Canada, 2012).

Efforts to develop emission inventories at the fine spatial and temporal resolution required

for urban-scale understanding of $CO_2$ emissions has been driven both by policy- and science-related
questions (Gurney et al., 2009; Patarasuk et al., 2016). From a policy perspective, improving $CO_2$
emission quantification is essential to independently evaluate whether anthropogenic mitigation
regulations are being effectively implemented. From a scientific perspective, gaining information
about anthropogenic $CO_2$ emissions from urban areas has been primarily motivated by atmospheric
$CO_2$ inversions, which are used to better understand the global carbon cycle (Gurney et al., 2009;
Patarasuk et al., 2016). Regardless of the motivation, quantification of $CO_2$ source/sink processes
currently uses two techniques: the bottom-up approach and the top-down approach. In the bottom-
up approach, local-scale activity level information is combined with appropriate emission factors to
infer emission rates. This method has been used widely to develop many inventories (EDGAR, 2010;
FFDAS, 2010; Gurney et al., 2009) but is limited by the accuracy of the input parameters. Conversely,
in the top-down approach, inverse modelling is used to exploit the variability in atmospheric mixing
ratios of $CO_2$ to identify the source/sink distributions and magnitudes; this method is limited by
insufficient mixing ratio data and uncertainties in simulating atmospheric transport (Pillai et al.,
2011). Given current policy needs, a strategy using solely bottom-up or top-down approaches is likely
insufficient to evaluate $CO_2$ emissions but rather a synthesis of the two methodologies is required
(Miller and Michalak, 2016). Successful examples of high-resolution $CO_2$ inventory development are
available on the urban scale, such as the Airparif inventory in Ile-de-France (publicly available at
http://www.airparif.asso.fr/en/index/index) and in Indianapolis, Los Angeles, Salt Lake City and
Phoenix through the Hestia project (Gurney et al., 2012), on the national scale, such as in China (Zhao
et al., 2012), and on the global scale (Wang et al., 2013). However, to our knowledge, there are



currently no studies that have quantified Canadian $CO_2$ emissions at the fine spatial and temporal
resolution required for urban analyses in Canada.

In an effort to address this gap, this study was focused on quantifying $CO_2$ emissions at a fine

spatial and temporal resolution in the GTA and southern Ontario (we expanded the inventory beyond
the urban area of the GTA so we could exploit information on $CO_2$ mixing ratios collected at rural
areas in central and south-western Ontario, proving additional sites for inventory validation). We
present the new high-resolution Southern Ontario $CO_2$ Emissions (SOCE) inventory, which quantifies
$CO_2$ emissions from seven source sectors (On-road, Off-road, Area, Point, Marine, Residential, and
Commercial natural gas combustion) at 2.5 km x 2.5 km spatial and hourly temporal resolution for
an area covering ∼26 % of the province of Ontario (∼$2.8 \times 10^5$ km$^2$). The SOCE inventory was used in
combination with the Environment and Climate Change Canada (ECCC) GEM-MACH chemistry-
transport model to simulate $CO_2$ mixing ratios in a domain including south-eastern Canada and the
northeastern USA (hereafter referred to as the "PanAm domain") for comparison with in situ
measurements made by the Southern Ontario Greenhouse Gas Network. Until now, estimates of
anthropogenic $CO_2$ emissions in the GTA were available only from the EDGAR v.4.2 (EDGAR, 2010)
and the FFDAS v2 (FFDAS, 2010) inventories, which have very different annual totals for this region
($1.36 \times 10^8$ vs. $1.05 \times 10^8$ tonnes $CO_2$, respectively). Therefore, we expect the results of this work will
improve our ability to quantify the emissions of $CO_2$ in the entire domain as well as the relative
contributions of different sectors, providing a more detailed characterization of the carbon budget in
the GTA.
**2.0 Methods**
*2.1 Geographic Domain*
The geographic focus of this study was the GTA in southern Ontario, Canada. The GTA is the largest
urban area in Canada; it comprises five municipalities, Toronto, Halton, Durham, Peel and York,





which together have a population exceeding 6 million (Statistics Canada., 2012b). Although the GTA
comprises only 0.07 % of Canadian land area, it represents over 17 % of the total population as a
result of rapid urbanization over the past few decades (Statistics Canada., 2012b). Therefore, high-
resolution characterization of $CO_2$ emissions can help integrate climate policy with urban planning.
This region is home to a network of measurement sites providing long-term, publicly available
datasets of atmospheric $CO_2$ mixing ratio measurements, *Sect. 2.2* (Environment Canada, 2011) which
can be used to evaluate model outputs and inventory estimates. In 2016 the government of Ontario
released a Climate Change Action Plan, which includes an endowment given to the Toronto
Atmospheric Fund of $17 million to invest in strategies to reduce greenhouse gas pollution in the
GTA (Ontario, 2016). Therefore this research can provide timely information on the carbon budget
in the GTA and help to implement effective reduction strategies.
*2.2 The Southern Ontario Greenhouse Gas Network*
Measurements of ambient $CO_2$ dry air mixing ratios began in 2005 in southern Ontario at the Egbert
station followed by the Downsview station (2007), Turkey Point station (2012) and Hanlan's Point
station (2014), Figure 1. Egbert is located ~75 km north-northwest of Toronto in a rural area,
Downsview is located ~20 km north of downtown core of the city of Toronto in a populated suburban
area, Turkey Point is located to the south-west of the GTA in a rural area on the north shore of Lake
Erie, and Hanlan's Point is located on Toronto Island, just south of the city of Toronto on the shore of
Lake Ontario. Site details and instrument types used can be found in Table 1. $CO_2$ measurements are
collected as a part of ECCC's Greenhouse Gas Observational Program. The measurement procedure
follows a set of established principles and protocols outlined by a number of international agencies
through recommendations of the *Meeting on Carbon Dioxide, Other Greenhouse Gases, and Related*
*Measurement Techniques,* coordinated by the World Meteorological Organization (WMO) every 2
years.





The atmospheric $CO_2$ observational program Egbert is based on non-dispersive infrared
(NDIR) methodology and fine-tuned for high precision measurements (Worthy et al., 2005).  A
detailed description of the NDIR observational system can be found in Worthy et al, (2005). The
atmospheric $CO_2$ observational programs at Turkey Point, Downsview, and Hanlan's Point are based
on Cavity Ring-Down Spectrometer (CRDS). Each Picarro CRDS system is calibrated in the ECCC
central calibration facility in Toronto before deployment to the field. The response function of the
analyzer is determined against 3 calibrated standards tanks (Low, Mid, High).  The working (W) and
target (T) tanks assigned to the system are also included in the injection sequence and calibrated.  At
each site, ambient measurements are made using two sample lines placed at the same level.  Each
sample line has separate dedicated sample pumps and driers ($\sim$ -30$^{o}$C).  Pressurized aluminum 30 L
gas cylinders are used for the working and target tanks.  The sample flow rate of the ambient and
standard tank gases is set at $\sim$300 cc/min. The injection sequence consists of a target and working
tanks sequentially passed through the analyzer for 10 minutes each every 2 days.  The ambient data
from line1 is passed through the analyzer for 18 hours followed by Line2 for 6 hours.   The
Line1/Line2 sequence repeats one time before the target and working tanks are again passed
through the system. The working and target tanks are calibrated on site at least once per year against
a single transfer standard transported between the sites and the central laboratory facility in
Toronto.   The $CO_2$ measurements from both the NDIR and CRDS analytical systems have a precision
of around 0.1 ppm based on one-minute averages and are accurate to within 0.2 ppm.
*2.3 GEM-MACH chemistry- transport model*
In this project, we used the GEM-MACH (Global Environmental Multi-scale–Modelling Air quality and
CHemistry) chemistry–transport model (CTM) (Gong et al., 2015; Moran et al., 2013; Pavlovic et al.,
2016; Talbot et al., 2008) to link surface emission estimates and atmospheric mixing ratios. GEM-
MACH is an on–line CTM embedded within the Canadian weather forecast model GEM (Côté et al.,



1998a; Côté et al., 1998b). The configuration of GEM-MACH used in our study has 62 vertical levels
from the surface to ~1.45 hPa on a terrain-following staggered vertical grid for a log-hydrostatic
pressure coordinate.  The thickness of the lowest layer was 40 m.  The PanAm domain used in our
simulations, which includes central and southern Ontario, as well as western Quebec and the
northeastern USA, is shown in Figure 1. The PanAm domain has 524 x 424 grid cells in the horizontal
on a rotated latitude-longitude grid with 2.5-km grid spacing and covers an area of approximately
1310 km x 1060 km (total domain area is $1.39 \times 10^6$ km$^2$).  A 24-hour forecasting period was used
with a 60-second time step for each integration cycle.  Meteorological fields (wind, temperature,
humidity, etc.) were re-initialized every 24 hours (i.e., after each model integration cycle); chemical
fields were carried forward from the previous integration cycle (i.e., perpetual forecast).  Hourly
meteorological and chemical boundary conditions were provided by the ECCC operational 10-km
GEM-MACH air quality forecast model (Moran et al., 2015).
In our study, we simulated two scenarios of $CO_2$ surface fluxes, indicated by the sum of the following:
Scenario 1:
• Anthropogenic fossil fuel $CO_2$ emissions within the province of Ontario estimated by the SOCE
inventory, available at 2.5 km x 2.5 km spatial and hourly temporal resolution, as described
in *Sect. 2.4*
• Anthropogenic fossil fuel $CO_2$ emissions estimated by the FFDAS v2 inventory (FFDAS, 2010)
outside of the province of Ontario (province of Quebec and USA), available at 0.1$^\circ$ x 0.1$^\circ$
spatial and hourly temporal resolution
• Biogenic $CO_2$ fluxes from the C-TESSEL land surface model, as described in *Sect. 2.5*




Scenario 2:
• Anthropogenic fossil fuel $CO_2$ emissions estimated by the FFDAS v2 inventory (FFDAS, 2010)

for the entire domain, available at 0.1° x 0.1° spatial and hourly temporal resolution

• Biogenic $CO_2$ fluxes from the C-TESSEL land surface model, as described in *Sect. 2.5*
$CO_2$ is not a usual chemical species considered by GEM-MACH but a set of special inert tracer fields
were added to GEM-MACH for this project to account for $CO_2$ concentration fields associated with
difference source sectors and the lateral boundaries.  The $CO_2$ boundary conditions set at the lateral
and top edges of the domain were obtained from the Monitoring Atmospheric Composition and
Climate (MACC) global inversion, v.10.2 (http://www.copernicus-atmosphere.eu/). Model simulated
specific humidity (q, kg/kg) was used to convert estimated $CO_2$ mixing ratios to dry air mixing ratios.
$CO_2$ dry air mixing ratios are hereafter referred to $CO_2$ mixing ratios.
*2.4 High-Resolution SOCE inventory development*
The high-resolution SOCE inventory was constructed primarily from a pre-existing carbon monoxide
(CO) inventory developed by the Pollutant Inventories and Reporting Division (PIRD) of ECCC as part
of the 2010 Canadian Air Pollutant Emissions Inventory (APEI). The CO inventory is a comprehensive
national anthropogenic inventory that includes emissions from area sources, point sources, on-road
mobile sources and off-road mobile sources, including aircraft, locomotive and marine emissions for
base year 2010 (Moran et al., 2015). This annual inventory at the provincial level compiled by PIRD
was transformed into model-ready emissions files using the Sparse Matrix Operator Kernel
Emissions (SMOKE, https://www.cmascenter.org/smoke/) emissions processing system for spatial
allocation (distribution of non-point source emissions to 2.5 km x 2.5 km (roughly 0.02° x 0.02°
resolution) using spatial surrogate fields) and temporal allocation (conversion of inventory annual
emission rates into hourly values) (Moran et al., 2015). More detailed information about the CO



inventory compilation and subsequent processing has been provided elsewhere (Environment
Canada, 2013; Moran et al., 2015; PIRD, 2016).

The objective of our work was to calculate $CO_2$ emissions based on this processed, model-

ready CO inventory for Ontario grid cells using sector-specific emission ratios estimated by the
Canadian National Inventory Report (NIR) (Environment Canada, 2012). Emission sources within
each sector of the CO inventory are classified by SCC (Source Classification Code) and were converted
to NFR (Nomenclature for Reporting) for accurate cross-reference with the NIR $CO_2$ and CO
estimates. Provincial totals for $CO_2$ and CO are estimated based on the NFR sources that are included
in the sector, producing the following sector-averaged $CO_2$:CO ratio:
$$CO_{2(sector,kt)} = CO_{(sector,kt)} * \frac{CO_{2(Ontario\ total,kt)}}{CO_{(Ontario\ total,kt)}} \qquad \text{Eq. (1)}$$
This sector-averaged $CO_2$:CO ratio is used to convert the APEI-based CO model-ready gridded
emissions fields into $CO_2$ emissions fields at the same spatial and temporal resolution. A detailed
outline of this conversion is presented for each sector in the following subsections. Unless otherwise
noted, temporal allocation of emissions in each sector is based on estimates made available by
SMOKE.
*2.4.1 Area emissions*
Area emissions are mostly small stationary sources that represent diffuse emissions that are not
inventoried at the facility level. In the APEI CO inventory, the major emission sources in the Area
sector include emissions from public electricity and heat production (1A1a), residential and
commercial plants (1A4a and 1A4b), stationary agriculture/forestry/fishing (1A4c), iron and steel
production (2C1), and pulp and paper (2D1). The NIR estimates an Ontario total from these (and
other minor sources) of 23,455 kt $CO_2$ and 218.8 kt CO, producing a $CO_2$:CO ratio of 107.2 kt $CO_2$/kt





CO. This ratio was applied to every Area sector grid cell belonging to Ontario in the domain to convert
sector CO emissions to $CO_2$ emissions.
*2.4.2 Point emissions*
Point emissions are stationary sources in which emissions exit through a stack or identified exhaust.
In the APEI CO inventory, the major emission sources in the Point sector include public electricity
and heat production (1A1a), stationary combustion in manufacturing industries and construction
(1A2f), chemical industry (2B5a), pulp and paper (2D1), iron and steel production (2C1) and other
metal production (2C5). Unlike the Area sector, we found that applying a single $CO_2$:CO ratio to every
facility did not produce realistic $CO_2$ emissions due to the significant variability in combustion
efficiency (and thus $CO_2$:CO ratio). Therefore, we used ECCC Facility Reported Data (Environment
Canada, 2015) to identify the geocoded location and annual average $CO_2$:CO for 48 individual facilities
in Ontario (Table S1) and applied the specific $CO_2$:CO ratios to the grid cells where the facilities were
located. In addition, stack height of individual facilities were included in the emission model to
optimize plume rise. All other point sources (minor facilities) were scaled by a sector average $CO_2$:CO
ratio of 313.1 kt $CO_2$/kt CO, calculated from Ontario total $CO_2$ and CO point-source emissions from
the NIR. Temporal allocation of emissions in the Point sector are based on facility level operating
schedule data collected by ECCC.
*2.4.3 On-road mobile emissions*
On-road emissions include the emissions from any on-road vehicles (quantified by the Statistics
Canada Canadian Vehicle Survey) (Environment Canada, 2013). In the APEI CO inventory, the major
emission sources in the On-road sector includes gasoline and diesel-powered light- and heavy-duty
vehicles (1A3b). The NIR estimates an Ontario total from these (and other minor on-road sources) of
44,458 kt $CO_2$ and 1508.3 kt CO, producing a $CO_2$:CO ratio of 29.5 kt $CO_2$/kt CO. This ratio was applied
to every On-road grid cell belonging to Ontario in the domain to convert sector CO emissions to $CO_2$.



Temporal allocation of emissions in the On-road sector is estimated using data collected in the FEVER
(Fast Evolution of Vehicle Emissions from Roadways) campaign in 2010 (Gordon et al., 2012a;
Gordon et al., 2012b; Zhang et al., 2012).
*2.4.4 Off-road mobile emissions*
Off-road emissions include the emissions from any off-road vehicles that do not travel on designated
roadways, including aircraft, all off-road engines, and locomotives. In the APEI CO inventory, the
major emission sources in the Off-road sector include civil aviation (1A3a), railways (1A3c), and
agriculture/forestry/fishing: off-road vehicles and other machinery (1A4c). Similar to the Point
sector, we found that applying a single $CO_2$:CO ratio to every grid cell did not produce realistic $CO_2$
emissions for the two airports in the GTA, Pearson International Airport (PIA) and Billy Bishop
Toronto City Airport (BBTCA). Therefore, we used air quality assessment reports compiled for each
airport (RWDI AIR Inc., 2009; RWDI AIR Inc., 2013) to identify the geocoded location and facility-
specific annual average $CO_2$:CO ratio. Sources of emissions from each airport include aircraft (landing
and take-off cycles), auxiliary power units, ground support equipment, roadways, airside vehicles,
parking lots, stationary sources and training fires; note that emissions from aircrafts in-transit
between airports, which are injected  in the free troposphere, have not been included in this
inventory (Moran et al., 2015; RWDI AIR Inc., 2009). Based on these two reports, we applied a ratio
of 175 kt $CO_2$/kt CO to the grid cell containing PIA and a ratio of 20 kt $CO_2$/kt CO to the grid cell
containing BBTCA. All other off-road sources belonging to Ontario grid cells were scaled by a sector
average $CO_2$:CO ratio of 7.2 kt $CO_2$/kt CO, calculated from NIR-reported Ontario total $CO_2$ and CO
emissions.
*2.4.5 Marine emissions*
Commercial marine emissions include the emissions from any marine vessels travelling on the Great
Lakes (quantified by the Statistics Canada *Shipping in Canada*) (Environment Canada, 2013). In the



APEI CO inventory, the major emission source in the Marine sector is national navigation (1A3d). The
NIR estimates an Ontario total from this source of 729.2 $CO_2$ and 0.86 kt CO, producing a $CO_2$:CO ratio
of 844.2 kt $CO_2$/kt CO. This ratio was applied to every marine grid cell in the domain to convert sector
CO emissions to $CO_2$.
*2.4.6 Residential and commercial emissions*
Residential and commercial $CO_2$ emissions reflect on-site combustion of natural gas for electricity
and heating, a source that we found was not included in the APEI CO inventory because of the high
efficiency of the furnaces and resulting low CO emissions. To include the $CO_2$ emissions from these
on-site furnaces, we used the Statistics Canada 2012 Report on Energy Supply and Demand to
quantify the amount of natural gas consumed by residential and commercial buildings in Ontario,
7969.6 gigalitres (Gl) and 4895.7 Gl respectively (Statistics Canada, 2012a). We used an emission
factor of 1879 g $CO_2$/m$^3$ natural gas combustion (Environment Canada, 2012) to estimate $CO_2$
emissions from residential and commercial on-site furnaces in Ontario to be 1.5 x 10$^7$ tonnes and 9.2
x 10$^6$ tonnes, respectively. These two emission totals were spatially allocated using a "capped-total
dwelling" spatial surrogate developed by ECCC and temporally allocated using the SMOKE emissions
processing system (Moran et al., 2015).
*2.5 Biogenic flux*
The net ecosystem exchange (NEE) fluxes used in our simulations were provided by the land surface
component of the ECMWF forecasting system, C-TESSEL (Bousetta et al., 2013). Fluxes are extracted
at the highest available resolutions, 15 x 15 km and 3 hour for January and February 2016 and 9 x 9
km and 3 hour for March. These data are interpolated in space and time to be consistent with our
model resolution. With our main priority being understanding anthropogenic emissions in the GTA,
we chose to analyze a period where the biogenic $CO_2$ flux is minimized and therefore this paper
focuses on three winter months in 2016, January to March inclusive.



### 3.0 Results and Discussion

#### 3.1 The SOCE inventory

Figure 1 displays the PanAm domain total anthropogenic $CO_2$ emissions estimated by the SOCE inventory for the province of Ontario portion (~0.02° x 0.02°) and by the FFDAS v2 inventory (0.1° x 0.1°) (FFDAS, 2010) for the remainder of the domain. Regions of high emissions typically correspond to population centers, for example the GTA in Ontario, Montreal and Quebec City in Quebec, and Chicago, Boston and New York City (amongst others) in the USA. Emissions from highways and major roadways are only clear in the province of Ontario (at higher spatial resolution) but industrial and large scale area sources are evident across the entire domain.

The total $CO_2$ emissions can be separated into contributions from the seven sectors in the province of Ontario described in *Sect. 2.4*. Figure 2 shows the anthropogenic $CO_2$ contributions from the Area sector, Residential and Commercial sector, Point sector, Marine sector, On-road sector and Off-road sector, focused on southern Ontario and the GTA. If we consider emissions from a domain including the area solely around the GTA (indicated by the black-box in Figure 2a), the total $CO_2$ emissions estimated by the SOCE inventory is 94.8 Mt $CO_2$ per year, Table 2. Figures 2a and b display the $CO_2$ emissions from the Area sector and from Residential and Commercial natural gas combustion in southern Ontario. These two sectors combined represent the largest source of $CO_2$ in the black-box area (41.6 Mt $CO_2$/year, contributing 43.9 % of the total). The majority of these emissions are concentrated in the GTA and surrounding urban areas as a result of a significant portion of the population (64 %) being reliant on natural gas for heat production (Statistics Canada, 2007; Statistics Canada, 2012a). Figure 2c represents emissions from the Point sector, contributing 24.4 Mt $CO_2$/year, 25.7 % of the total. The largest point source emitters are located on the western shore of Lake Ontario (Hamilton and surrounding areas) as this area is one of the most industrialized regions of the country with intensive metal production activities. Figures 2d, e and f display $CO_2$ emissions from various



transportation sectors, Marine, On-road, and Off-road respectively, which together contribute more
than 30 % of total $CO_2$ emissions in the area within the black box. While emissions from marine
activity are minimal, those from On-road and Off-road sources are significant (25.0 % and 5.3 %,
respectively), concentrating on the major highways connecting the various population centres of the
GTA to the downtown core, as well as at PIA located within the city.
*3.2 Comparison of the SOCE inventory with other inventories*
The EDGAR v4.2 inventory estimates $CO_2$ emissions on an annual basis and by sector based on
Selected Nomenclature for Air Pollution (SNAP) sub-sectors while FFDAS v2 provides hourly mean
grid cell totals. Table 2 shows a comparison between the sectoral $CO_2$ estimates of the SOCE and
EDGAR v4.2 inventories (SNAP sectors were grouped to correspond to SOCE sectors, Table S2) as
well as the domain total estimated by the FFDAS v2 inventory for the area surrounding the GTA (the
black-box area outlined in Figure 2a). There is a significant discrepancy between the $CO_2$ emissions
estimated by the SOCE and EDGAR v4.2, inventories both in the relative sectoral contributions as well
as domain total (percent difference >35 %). The largest sectoral discrepancies are in the Point and
the On-road sectors, where the EDGAR v4.2 inventory estimates a contribution 1.9 and 1.7 times
larger than that of the SOCE inventory, respectively. Figure 3 shows a comparison of the spatial
distribution of the $CO_2$ inventory predicted by a) FFDAS v2, b) EDGAR v4.2, and c) SOCE (aggregated
to 0.1º x 0.1º to match the resolution of EDGAR v4.2 and FFDAS v2) for the GTA area. Figure 3 reveals
that the largest differences between the SOCE inventory and the EDGAR v4.2 inventory is the $CO_2$
emissions in the GTA; EDGAR v4.2 predicts much higher emissions in the GTA (in some grid cells,
differences are an order of magnitude), particularly in the downtown core relative to the SOCE
inventory.
Although there is no sectoral breakdown in the FFDAS v2 inventory, the domain total around
the GTA can be compared to that of the SOCE inventory, Table 2. Unlike the comparison with the



EDGAR v4.2 inventory, there is a closer agreement between the FFDAS v2 inventory and the SOCE
inventory (difference of ~10 %). The comparison plots in Figure 3 show a good agreement of the
spatial variability of emissions in the GTA between the FFDAS v2 and SOCE inventories; however, the
gradient between urban and rural areas is not as sharp in the SOCE inventory as it is in the FFDAS v2
inventory. Furthermore, emissions along the western shore of Lake Ontario (Hamilton and the
surrounding areas) are predicted to be larger in the SOCE inventory relative to FFDAS v2. The FFDAS
v2 inventory was interpolated to $0.02°$ x $0.02°$ using a mass conservative interpolation scheme to
allow the production of a difference plot of the two inventories, SOCE minus FFDAS v2, shown in
Figure S1. The difference plot reveals the largest divergence between the inventories occurs in the
GTA and Ottawa, with the FFDAS v2 inventory estimating >1000 g $CO_2$/second (~30 kt $CO_2$/year)
more than the SOCE inventory in some grid cells. In addition to similar spatial variability, the FFDAS
v2 and SOCE inventories also have similar temporal variability. Figure S2 shows the diurnal profile
of estimated emissions from January-March for both the FFDAS v2 and SOCE inventories for the
black-box area in the PanAm domain. Both inventories allocate the highest emissions between 08:00
and 18:00 and the lowest emission between 00:00 and 5:00, however the amplitude of the diel cycle
is higher in SOCE, and emissions in the morning are as high as in the afternoon. FFDAS allocates a
relatively larger proportion of the emissions to the 15:00 – 19:00 period.
*3.3 Preliminary analyses using the SOCE, FFDAS v2 and EDGAR v4.2 inventories with FLEXPART*
To investigate the impact of the differing inventories on ambient mixing ratios, preliminary analyses
were run with footprints generated by the FLEXPART driven by GEM meteorology and products were
compared against the measured $CO_2$ gradient between the Downsview and TAO (43.7°N, 79.4°W, a
temporary site decommissioned in January 2016) stations in the year 2014. Observed gradients
ranged from +20 to -10 ppm. Figure S3 displays the measured and modelled $CO_2$ gradients.  These
results show that when the EDGAR v4.2 inventory was used, simulated $CO_2$ gradients were



consistently overestimated by ~10-60 ppm relative to observations. Conversely, when the SOCE
inventory was used, a higher level of agreement was obtained between simulated mixing ratios and
measurements; however, none of the model simulations were able to capture times when the
gradient was negative ($CO_{2,TAO}$ > $CO_{2,Downsview}$), an effect we believe to be due to the TAO inlet being
~60 m above ground level and surrounded by many high-rise buildings creating canyon flows and
turbulence which are not properly accounted for in GEM at this resolution.  These factors contributed
to the decommissioning of TAO in January 2016. The poor performance of our model system when
using the EDGAR v4.2 inventory to simulate $CO_2$ mixing ratios was also found by a study quantifying
on-road $CO_2$ emissions in Massachusetts, USA (Gately et al., 2013). In this study, EDGAR emission
estimates were found to be significantly larger than any other inventory by as much as 9.3 million
tons, or >33 %. The difference in estimates between the EDGAR v4.2 and the SOCE inventories is
likely explained by their underlying differences in methodology. Being a global product and not
specifically designed for mesoscale applications, the EDGAR v4.2 inventory estimates $CO_2$ emissions
based on country-specific activity data and emission factors, however the spatial proxies used to
disaggregate the data are not always optimal. A study performed by McDonald et al. (2014) showed
that the use of road density as a spatial proxy for vehicle emissions in EDGAR v4.2 causes an
overestimation of emissions in population centers (McDonald et al., 2014). Given the much larger
emission estimates for On-road $CO_2$ from EDGAR v4.2 (Table 2), this also seems to be an issue in the
GTA. Based on this large discrepancy, the EDGAR v4.2 inventory was not further used in this study
and we focussed on the inventories developed for regional scale studies.

When similar preliminary analyses were run with FLEXPART footprints using the FFDAS v2

inventory, Figure S3, good agreement was observed with $CO_2$ gradients measured between the
Downsview and TAO stations. We are confident that the enhanced measurement agreement between
the FFDAS v2 and SOCE relative to EDGAR v4.2 is due to improved methodology; spatial allocation of
emissions in FFDAS v2 is achieved through the use of satellite observations of nightlights from human



settlements from the U.S. Defense Meteorological Satellite Program Operational Linescan System
(DMSP-OLS).

Beyond the differences in methodology for estimating and allocating emissions, it is

important to note that the emissions reported in Table 2 by the FFDAS v2, SOCE and EDGAR v4.2
inventories also fundamentally differ in time period quantified. The emissions reported for both
FFDAS v2 and the SOCE are based on emissions from three winter months (January-March 2010)
extrapolated for the entire year. However, emissions from EDGAR v4.2 are annual averages of all
twelve months of 2010. Since $CO_2$ emissions in the GTA are higher in the winter months relative to
the summer months because of increased building and home heating, it is likely that the average
annual estimates of SOCE and FFDAS v2 are slightly overestimated. This does not affect the relative
agreement between SOCE and FFDAS v2 however it does further increase the divergence between
the EDGAR v4.2 and SOCE and FFDAS v2 inventories.  Following this and the improved agreement
with observations, the FFDAS v2 inventory was used with the SOCE inventory for all subsequent
modelling analyses.
*3.4 Simulation of CO₂ mixing ratios in the Greater Toronto Area*
We used the GEM-MACH CTM and the SOCE and FFDAS v2 inventories to simulate hourly $CO_2$ mixing
ratios in the PanAm domain. The model framework was evaluated for a continuous three-month
period, January-March 2016 using four sampling locations in the GTA, Figure 1 (note that
measurements for the Hanlan's Point site were not available until January 14, 2016). Figure 4
displays afternoon (12:00-16:00 EST) measured and simulated $CO_2$ mixing ratios produced with the
SOCE and FFDAS v2 inventories for the two emissions scenarios described in *Sect. 2.3* for the month
of February (Figures S4 and S5 show the same figure for other months). We chose to present only
afternoon data as this is the time of day when the mixed layer is likely to be the most well-developed;
nighttime and morning data showed largest variations in observations as a result of the shallow



boundary layer causing surface emissions to accumulate within the lowest atmospheric layers
(Breon et al., 2015; Chan et al., 2008; Gerbig et al., 2008). During the night, atmospheric mixing ratios
are most sensitive to vertical mixing, an atmospheric process that is difficult to model for stable
boundary layers.
The time series comparisons at all four sites demonstrate the model's general ability to
capture variability in observations of $CO_2$, albeit with better skill for the Downsview and Egbert sites
(this is particularly clear when we look at model-measurement difference plots, Figure S6). The
model is able to capture many extreme events of mixing ratio increases and decreases, such as
February 11-14, 2016 at the Downsview site; however, some short time periods are poorly simulated,
such as January 21-23, 2016 at Hanlan's Point, when the model significantly overestimated measured
$CO_2$. Generally, mixing ratios simulated by the FFDAS v2 inventory are similar or larger than those
produced when the SOCE inventory is used, with differences most noticeable at the Downsview and
Hanlan's Point sites. This was expected as the difference plot shown in Figure S1 reveals that the
SOCE and FFDAS v2 inventories diverge the most in the GTA (where the Downsview and Hanlan's
Point sites are located) and are more similar in rural areas (where the Turkey Point and Egbert sites
are located).
Measured $CO_2$ mixing ratios have a typical diurnal pattern, in which mixing ratios are higher
at night and lower during the day, despite higher emissions during the day. This results from the daily
cycle of the mixed layer, which is shallow at night due to thermal stratification and deepens during
the day due to solar heating of the surface. Figure 5 displays the measured and modelled mean
diurnal profile of $CO_2$ at the four sites in our network using data from January-March, 2016 (note
difference in y-axis scale for urban vs. rural sites). At all four sites, the shapes of the modelled and
measured mixing ratios throughout the day agree very well, suggesting that the GEM meteorology in
our framework is capturing the diurnal variation in emissions and the boundary layer evolution. At



the Downsview site, there is a very strong agreement between the modelled and measured diurnal
profiles when using the SOCE inventory, whereas the FFDAS v2 simulated profile largely
overestimates mixing ratios, particularly at nighttime. This is consistent with the FFDAS inventory
having larger emissions than the SOCE inventory during the night (Fig S2). At the Hanlan's Point site,
a difference of $\sim$ 5 ppm $CO_2$ is observed when using the SOCE inventory relative to measurements;
however, similar to the Downsview site, the FFDAS v2 simulated profile has a larger difference of
$\sim$10 ppm $CO_2$. At both the Egbert and Turkey Point sites, the use of both inventories similarly
overestimates the diurnal pattern of $CO_2$ mixing ratios by $\sim$3-5 ppm, again likely a result of the
similarities of these two inventories at these sites, Figure S1. At all four sites, it is possible that some
of the biases that are observed in simulated $CO_2$ mixing ratios may arise from inaccuracies in the
boundary $CO_2$ provided by MACC; this aspect was not, however, further explored in this study.
*3.5 Quantifying model-measurement agreement*
Figure 6 shows scatter plots of afternoon (12:00-16:00 EST) modelled versus measured $CO_2$ mixing
ratios from January- March, 2016 at the four sites used in this study. The top row shows the
correlation between measured and modelled mixing ratios using the SOCE inventory and the bottom
row shows the correlation using the FFDAS v2 inventory. It is immediately clear that there is a
stronger model-measurement correlation at the Downsview and Egbert sites (R > 0.75) relative to
that of Hanlan's Point or Turkey Point (R < 0.53). The difficulty with accurately simulating $CO_2$ mixing
ratios at Hanlan's Point and Turkey Point may arise from their proximity to shorelines, Hanlan's Point
to Lake Ontario and Turkey Point to Lake Erie (see Figure 1). Differential heating of land versus water
near these lakes creates pressure gradients driving unique circulation patterns (Burrows, 1991; Sills
et al., 2011). These circulation patterns are difficult for models to capture and therefore may
contribute to the relatively poor correlation observed at Hanlan's Point and Turkey Point.





It is also clear from Figure 6 that simulating $CO_2$ mixing ratios at the Egbert and Turkey Point
sites using either the FFDAS v2 or the SOCE inventory results in similar performance, likely because
the emissions estimated by the two inventories are similar in the vicinity of these two rural sites (see
also Figure 5). However at both the Downsview and Hanlan's Point sites, using the SOCE inventory
provided a slightly higher correlation and reduced RMSE and MB relative to using the FFDAS v2
inventory. The improvement by using the SOCE inventory is likely a result of both the improved
spatial resolution (2.5 km versus 10 km), and therefore more accurate allocation of emissions to grid
cells, and also a better estimation of emission magnitudes, as large differences are shown in Figures
3 and S1.
*3.6 Sectoral contributions to simulated $CO_2$ mixing ratios*
One of the major advantages of the SOCE inventory over the FFDAS v2 inventory is the availability of
sectoral emission estimates. Figure 7 displays the sectoral percent contributions to diurnal $CO_2$
mixing ratio enhancements (calculated as local $CO_2$ mixing ratios above the MACC estimated
background) for the Downsview station in February 2016 averaged by the day of week (Figures S7
and S8 displays the same for other months). This figure clearly demonstrates the importance of Area
emissions (defined here as the sum of the Area + Residential natural gas combustion + Commercial
natural gas combustion) to simulated $CO_2$ mixing ratios, reaching ~80 % contribution in the early
morning and late evening, consistent with times when emissions from home heating are the
dominant source of $CO_2$. Contributions from Area emissions decrease to ~35 % midday, which
coincides with when emissions from other sources, such as On-road, gain importance. In the midday,
emissions from the On-road sector can contribute ~50 %, which is consistent with transportation
patterns of the times when the population is travelling to and from work and other activities. The
relative contributions to $CO_2$ mixing ratios from point source emissions is quite variable during the
course of a day and week, but generally seems to increase in the early morning and evening and can




contribute a significant portion of total $CO_2$ emissions (up to ~20 %). Figure 7 indicates that biogenic
sources of $CO_2$ play a negligible role during January-March in the GTA. Recent studies, however, have
shown the importance of the biospheric contribution (up to ~132-308 g $CO_2$ $km^{-2}$ $s^{-1}$) to measured
$CO_2$ in urban environments during the growing season (Decina et al., 2016). Therefore, this finding
supports the importance of modelling $CO_2$ in the wintertime in cities like the GTA to avoid
complications associated with biospheric contributions. The new ability to understand the sectoral
contributions to $CO_2$ mixing ratios in the GTA and southern Ontario has implications from a policy
perspective; with recent initiatives to curb $CO_2$ emissions, understanding from which sector the $CO_2$
is being emitted could be useful to assess how effective applied mitigation efforts have been or where
to target future efforts. These efforts could be complemented by running simulations with additional
tracers, such as carbon monoxide (CO), nitrogen oxides ($NO_x$), or stable carbon isotopes ($^{12}C$ and $^{13}C$)
to gain further insight.
*4.0 Conclusions*
We presented the SOCE inventory for southern Ontario and the GTA, the first, to our knowledge, high-
resolution $CO_2$ inventory for southern Ontario and for a Canadian metropolitan region. The SOCE
inventory provides $CO_2$ emissions estimates at 2.5 km x 2.5 km spatial and hourly temporal
resolution for seven sectors: Area, Residential natural gas combustion, Commercial natural gas
combustion, Point, Marine, On-road and Off-road. When compared against two existing $CO_2$
inventories available for southern Ontario, the EDGAR v4.2 and the FFDAS v2 inventories, using
FLEXPART footprints, the SOCE inventory had improved model-measurement agreement; FFDAS v2
agreed well with in situ measurements, but the EDGAR v4.2 inventory systematically overestimated
mixing ratios. We developed a model framework using the GEM-MACH chemistry-transport model
on a high-resolution 2.5 km x 2.5 km grid coupled to the SOCE and FFDAS v2 inventories for
anthropogenic $CO_2$ emissions and C-TESSEL for biogenic $CO_2$ fluxes. We compared output simulations



to observations made at four stations across southern Ontario and for three winter months, January
– March, 2016. Model-measurement agreement was strong in the afternoon using both
anthropogenic inventories, particularly at the Downsview and Egbert sites. Difficulty in capturing
mixing ratios at the Hanlan's Point and Turkey Point sites was hypothesized to be a result of their
close proximity to shorelines (Lake Ontario and Lake Erie, respectively) and the model's inability to
capture the unique circulation patterns that occur in those environments. Generally, across all
stations and months, simulations using the SOCE inventory resulted in higher model-measurement
agreement than those using the FFDAS v2 inventory, quantified using R, RMSE and mean bias. In
addition to improved agreement, the primary advantage of the SOCE inventory over the FFDAS v2
inventory is the sectoral breakdown of emissions; using average day of week diurnal mixing ratio
enhancements, we were able to demonstrate that emissions from area sources can contribute >80 %
of $CO_2$ mixing ratio enhancements in the early morning and evening with on-road sources
contributing >50 % midday. The applications of the SOCE inventory will likely show future utility in
understanding the impacts of $CO_2$ reduction efforts in southern Ontario and identify target areas
requiring further improvement.
*Author Contributions*
The SOCE inventory was prepared by Stephanie C. Pugliese, with critical input from Felix Vogel and
Jennifer Murphy. The CO inventory which the SOCE inventory is based upon was provided by Mike
Moran, Junhua Zhang and Qiong Zheng. The GEM-MACH modelling analyses were performed by
Shuzhan Ren and Craig Stroud. The ambient $CO_2$ data were collected by Douglas Worthy and his team
at Environment and Climate Change Canada. The MACC and C-TESSEL products used in our model
simulations were provided by Gregoire Broquet. The data was analyzed and interpreted for
publication by Stephanie C. Pugliese. This manuscript was written by Stephanie C. Pugliese, with
critical input from Jennifer Murphy, Felix Vogel and Mike Moran.



*Acknowledgements*
The authors are thankful to Robert Kessler, Michelle Ernst, Lauriant Giroux, Senen Racki and Lin
Huang for their efforts collecting the $^{12}CO_2$ and $^{13}CO_2$ measurements at Environment and Climate
Change Canada. They would also like to thank Elton Chan for providing the FLEXPART footprints and
for Pegah Baratzadeh for help creating the SOCE inventory.





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





**Table 1**: **Summary of atmospheric measurement programs in Southern Canada operated by**
**Environment and Climate Change Canada**

| Start Date | Site Name | Coordinates | Elevation (asl) | Intake Height | In-situ Instrumentation |
|---|---|---|---|---|---|
| March, 2005 | Egbert | 44.231037N, 79.783834W | 251m | 3m, 25m* | NDIR |
| November, 2010 | Downsview | 43.780491N, 79.468010W | 198m | 20m | NDIR |
| November, 2012 | Turkey Point | 42.635368N, 80.557659W | 231m | 35m | CRDS |
| June, 2014 | Hanlan's Point | 43.612201N 79.388705W | 87m | 10m | CRDS |

* At Egbert, a 25 m tower was installed in March 9, 2009
NDIR = Non-dispersive infrared
CRDS = cavity ring-down spectroscopy




**Table 2: Anthropogenic CO₂ emissions for the year 2010 in the black-box area (shown in**
**Figure 2a) by sector. Values in parentheses indicate the percentage contribution of the**
**sector to the total CO₂ emissions in the black-box area.**

| Sector | FFDAS v2‡ CO₂ Inventory (Mt CO₂/year) | EDGAR v4.2# CO₂ Inventory (Mt CO₂/year) | SOCE CO₂ Inventory (Mt CO₂/year) |
|---|---|---|---|
| **Area*** | - | 46.2 (33.9 %) | 41.6 (43.9 %) |
| **Point** | - | 45.9 (33.7 %) | 24.4 (25.7 %) |
| **Marine** | - | 0.12 (0.10 %) | 0.10 (0.10 %) |
| **On-road** | - | 41.2 (30.2 %) | 23.7 (25.0 %) |
| **Off-road** | - | 2.95 (2.2 %) | 5.01 (5.3 %) |
| **Total** | 104.8 | 136.4 | 94.8 |

*Area sector represents the summation of Area + Residential + Commercial natural gas combustion.
#The EDGAR inventory v4.2 can be found at http://edgar.jrc.ec.europa.eu.
‡The FFDAS v2 inventory can be found at http://hpcg.purdue.edu/FFDAS/.






















**Figure 1: Total anthropogenic CO$_2$ emissions for a weekday in February 2010 estimated by the SOCE inventory for the province of Ontario and by the FFDAS v2 inventory for the remainder of the GEM-MACH PanAm domain. Locations of in-situ measurements of CO$_2$ in the Southern Ontario GHG Network are shown in the inset (Downsview = square, Egbert = circle, Hanlan's Point = triangle, Turkey Point = diamond). The Downsview and Hanlan's Point sites are both located in the GTA. Units: g CO$_2$/second/grid cell.**





**Figure 2: Anthropogenic CO$_2$ emissions for a weekday in February 2010 in southern Ontario.**
**Emissions are estimated by the SOCE inventory for the (a) Area sector; (b) sum of the**
**Residential and Commercial sectors; (c) Point sector; (d) Marine sector; (e) On-road sector;**
**(f) Off-road sector. Units: log$_{10}$(g CO$_2$/second/grid cell).**





748 FFDAS v2 Domain Total:
    1.05x10$^8$ tonne/year

EDGAR v4.2 Domain Total:
1.36x10$^8$ tonne/year

SOCE Domain Total:
9.48x10$^7$ tonne/year

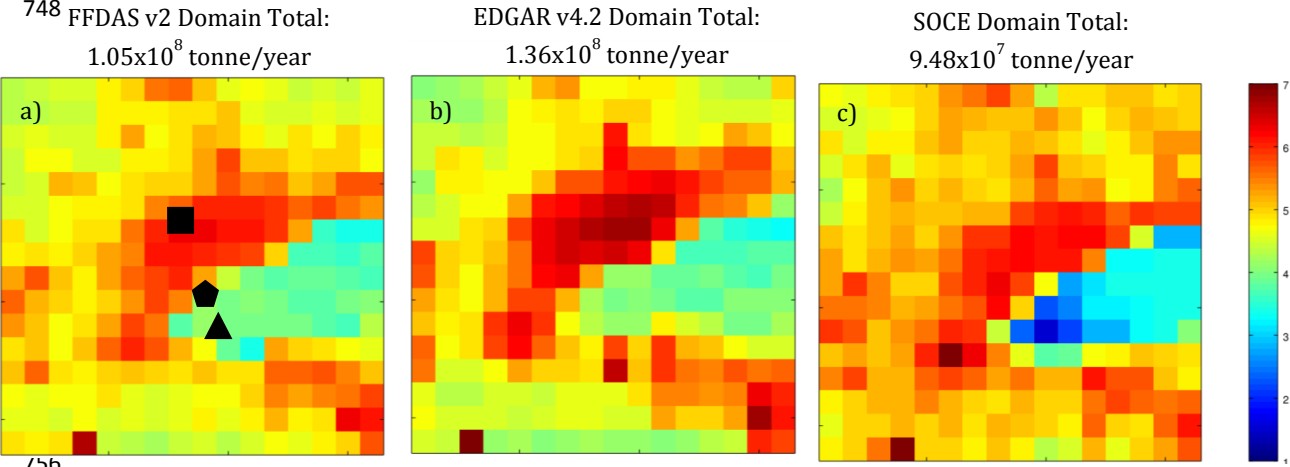

756

**Figure 3: Comparison of spatial distribution of annual CO$_2$ emissions inventories for the black-box area (shown in Figure 2a) at 0.1° x 0.1° resolution. Panel a) shows the FFDAS v2 inventory estimate, Panel b) shows the EDGAR v4.2 inventory estimate and Panel c) shows the SOCE inventory estimate. Units: log$_{10}$(tonne CO$_2$/year/grid cell). Domain totals are shown on top of each panel and locations of in-situ measurements of CO$_2$ for three stations in the Southern Ontario GHG Network are shown on Panel a (Downsview = square, Hanlan's Point = triangle, TAO = pentagon). The other two stations, Egbert and Turkey Point, are located outside this area.**

765





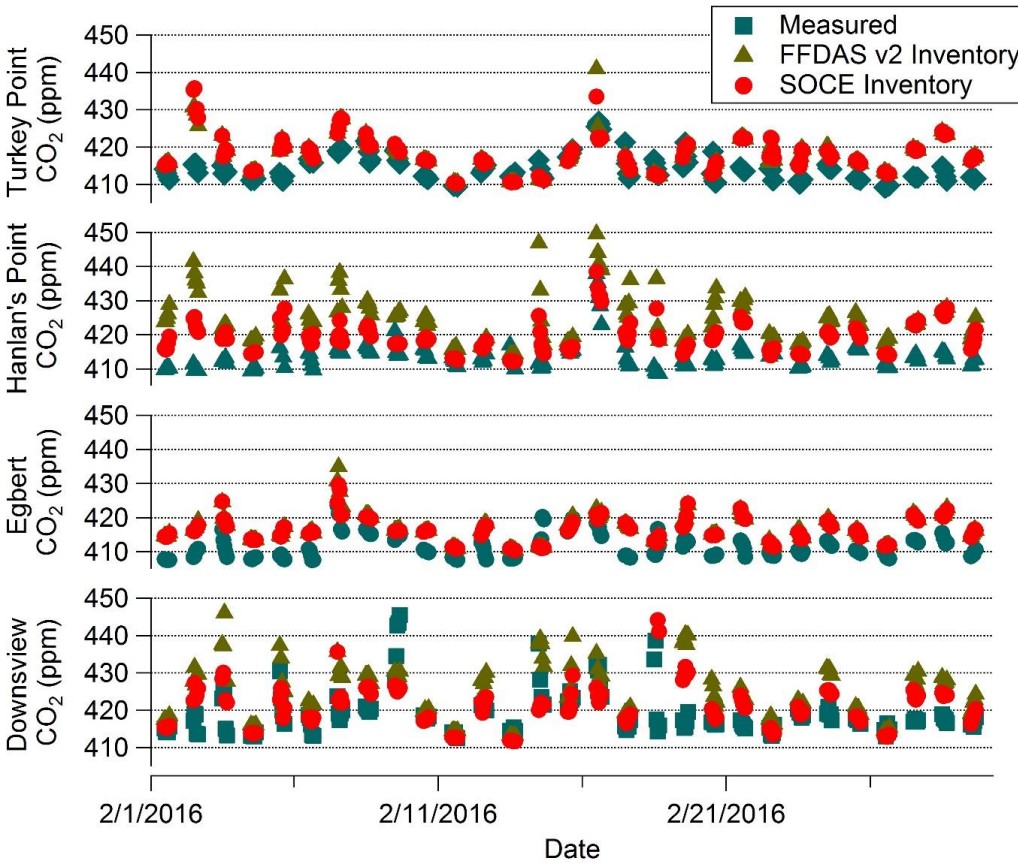

766

**Figure 4: Time series of measured (blue) and modelled February afternoon (12:00-16:00 EST)**
**CO₂ mixing ratios for the four sites used in this study. The red and gold markers are the**
**modelled mixing ratios when using the SOCE CO₂ inventory and the FFDAS v2 inventory,**
**respectively.**






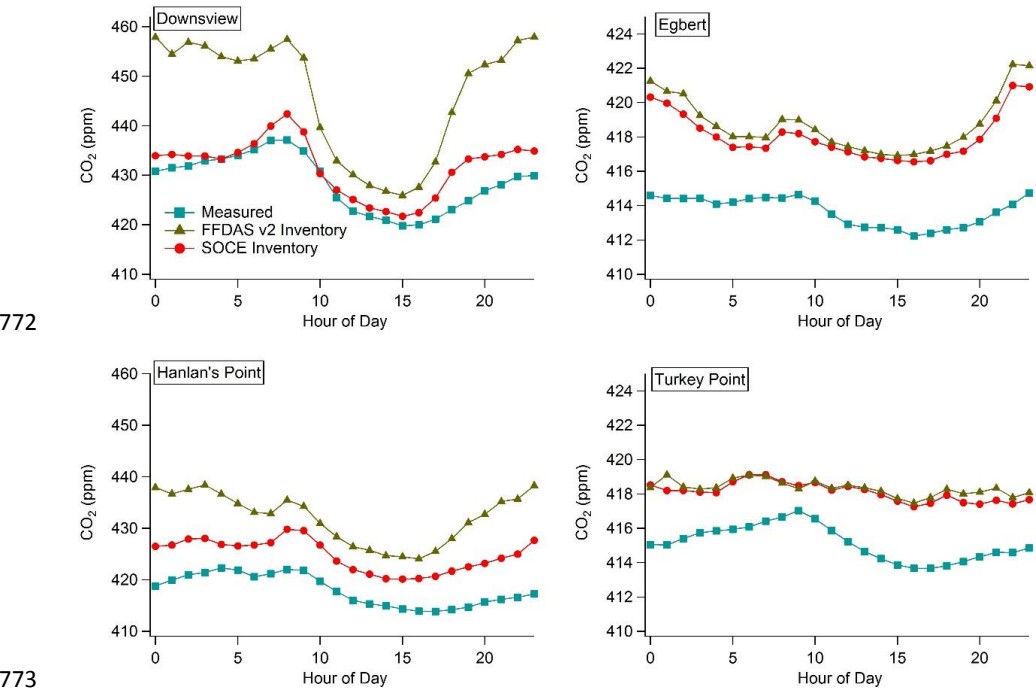




**Figure 5: Time series of mean measured (blue) and modelled diurnal $CO_2$ mixing ratios at the four sites considered in this study for January – March 2016. The red and gold markers are the modelled diurnal mixing ratios when using the SOCE $CO_2$ inventory and the FFDAS v2 inventory, respectively. Note the difference in scale for urban and rural sites.**

779





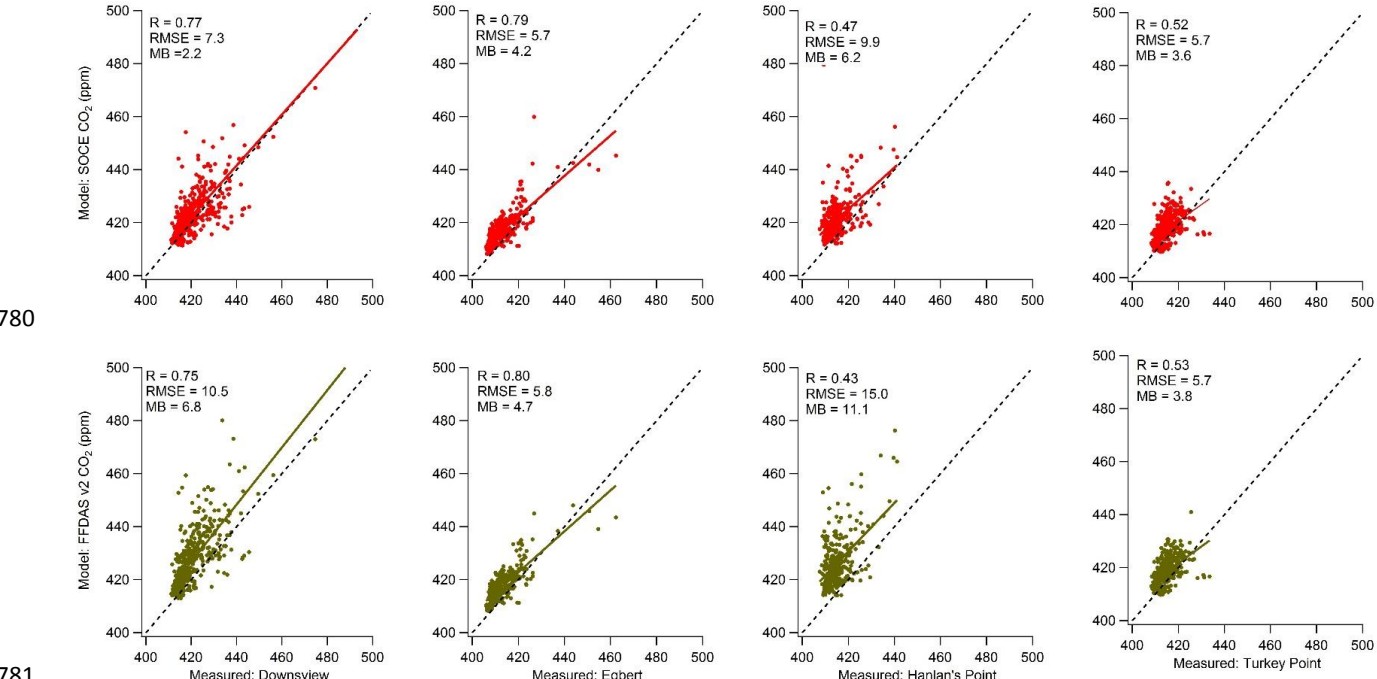

**Figure 6: Scatter plot of the modelled and measured afternoon (12:00-16:00 EST) CO$_2$ mixing ratios from January-March, 2016 at the four monitoring stations used in this study. The top and bottom panels show measurement-model correlation when the SOCE inventory and the FFDAS v2 inventory were used, respectively. The model vs. measurement Correlation Coefficient (R), root mean square error (RMSE) and mean bias (MB) (units: ppm) are provided within each panel. Solid lines are the standard major axis regression lines and dashed lines are 1:1 lines shown for reference.**



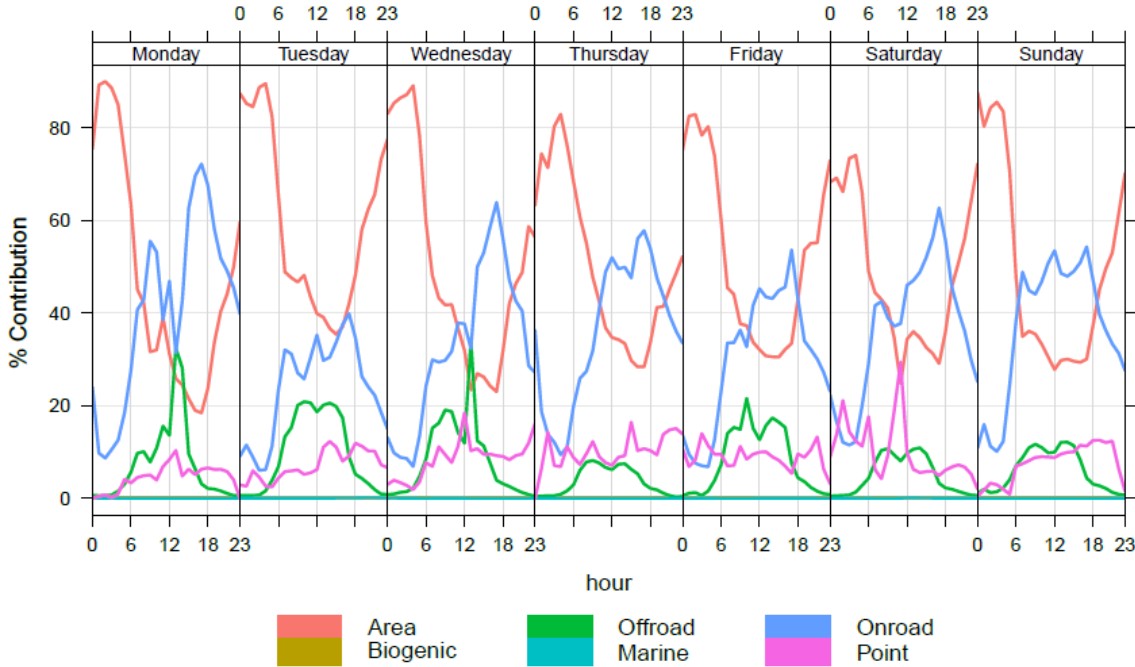

**Figure 7: Modelled sectoral percent contributions to diurnal local $CO_2$ enhancement for February 2016 at Downsview averaged by day of week. Note: Area = Area + Residential natural gas combustion + Commercial natural gas combustion. (Time zone is EST).**