# Peer review of "High-resolution quantification of atmospheric CO₂ mixing ratios in the Greater Toronto Area, Canada"

_Atmospheric Chemistry and Physics, 2017_

## Referee Comment (RC1) · Anonymous Referee #1 · 3 Sep 2017

The authors develop a high-resolution CO2 emissions inventory (SOCE) for the Toronto region, and evaluate it against in ambient CO2 measurement. The SOCE inventory appears to perform well against ambient observations, along with FFDAS, a commonly used CO2 inventory. The main advance of this study is sectoral information provided that FFDAS does not, which is significant. The sectoral information is useful to cities in their efforts to mitigate greenhouse gas emissions. Also, not many cities have urban CO2 monitoring networks from which to evaluate emission inventories. This study provides a useful framework by which to perform evaluations in the future.

Overall, I found the manuscript to be well written, figures to be clear, and findings well

supported. I do have some questions about the methods used to construct the bottom-up inventory, especially with the use of carbon monoxide (CO) to scale up to CO2 emissions. Most of the rest of my comments are minor in nature and relate to clarity. My recommendation is that this manuscript be published with a moderate amount of revisions.

General Comments

1. As I understand, there is provincial-level estimates of CO and CO2 that are reported by the Canadian National Inventory Report (NIR), with broad sectoral information (e.g., area, point, on-road, off-road). Then there is also a gridded inventory of CO emissions that is processed at 2.5 km x 2.5 km. The authors' reconcile the provincial-level CO2/CO estimates with the gridded CO inventory to arrive at a gridded CO2 inventory.

My main critique with this approach is with the CO2/CO emission ratios reported. Some of these values seem unbelievable, which reflect potential problems in either CO2 or CO emissions reported by the NIR (more likely from CO). For example, the on-road CO2/CO emissions ratio reported here (Line 247) is 29.5 g CO2/g CO. This is equivalent to a CO emission factor of 33.9 g CO/kg CO2 or 110 g CO/kg fuel (using a carbon fraction of 0.85 g C/g fuel for gasoline). Roadway studies report tailpipe CO emission factors from gasoline cars at 10-20 g/kg fuel [McDonald et al., 2013]. The factors reported here appear too high. Also, based on the point source emission factor of 313.1 g CO2/g CO (Line 239), ∼0.5% of the carbon emitted is as CO and the rest from CO2. This is a very small number in the denominator from which to scale to CO2 emissions, introducing potentially large uncertainties in industrial CO2 emissions.

Ultimately, I found the reporting of CO2/CO emission ratios distracting and not central to the inventory constructed. What I believe the authors' are really doing here is using the gridded CO inventory as spatial/temporal proxies for CO2 emissions, and downscaling CO2 emissions from the provincial-level to grid cells by sector. Rather than report CO2/CO emission ratios that are dubious, I suggest reframing inventory

methods to emphasize the use of CO as spatial/temporal proxies for CO2.

2. In Section 2.4, too many significant figures are reported in emission inventory estimates, which suggest a high degree of certainty in emissions that is unwarranted (especially for CO). Suggest using 2-3 significant figures at most.

Specific Comments

3. Line 207. How are CO2 emissions from the Canadian National Inventory Report (NIR) estimated? Are these based on energy or fuel use statistics, or from engineering calculations? Since CO2 emissions is the focus of this paper, it would be helpful to include a few sentences on the basis for how CO2 emissions are estimated in national reporting.

4. Equation 1. The "total" subscript was confusing to me. I believe what the authors' mean is "sector" in the third term of the equation (e.g., total area, point source, on-road, etc.) and "sub-sector" in the first two terms.

5. Section 2.4.1. Typically when I think of area emissions they are dispersive sources that include residential, commercial, AND industrial sources. Suggest that this source category be renamed to something like "Area industrial emissions".

6. Lines 232-234. The combustion efficiency is not actually that variable, it is just that CO emissions are almost negligible from point sources and hence why the CO2:CO ratio is variable due to a tiny denominator (see Comment 1). Suggest re-wording of this sentence.

7. Lines 255-256. Lawn equipment and other small two- and four-stroke gasoline engines (e.g., snow equipment) have been shown to be a significant source of CO emissions [Gordon et al, 2013; Volckens et al., 2007; Bishop et al., 2001]. Where would they show up in the APEI, or are they included here? More importantly, how are off-road gasoline engines specifically accounted for in this study, which contribute high amounts of CO, but consume small amounts of fuel? Other off-road diesel equipment would

consume significantly larger amounts of fuel than off-road gasoline engines, but have much lower CO emission factors. Not properly accounting for off-road emissions of CO between gasoline and diesel engines could affect the scaling of off-road emissions of CO to CO2.

8. Lines 363-365. Some more description of the FLEXPART model is needed here, or could warrant a short paragraph in the methods section. Specifically, how many hours was the back trajectory simulated for? Was this run for each site? I'm guessing the emission inventories are then multiplied by the footprint to arrive at concentrations, and then compared with ambient CO2 measurements. Also, a reference to FLEXPART and some description of the model would be helpful to a reader unfamiliar with the model.

9. Line 365. This appears to be the first mention of the TAO site. It would be helpful to describe this location in more detail in Section 2.2. Also, it is not clear why looking at gradients between Downsview and TAO is a useful metric. Is it because TAO is a downwind site, whereas Downsview is downtown? In general, for a reader unfamiliar with Toronto, it would be helpful to describe locations as urban or rural more explicitly throughout Results and Discussion.

10. Lines 492-494. Focusing on wintertime months, while easier from a modeling perspective, would bias our understanding of CO2 emissions towards wintertime sources. The sources and spatial patterns of emissions vary between winter and summer. For example, peaking plants could be important in summertime [Farkas et al., 2016]. Seems like we should try to understand both periods.

11. Figure 5. This plot could benefit from some uncertainty bands on the CO2 measurements, such as the standard deviation or 95% confidence interval of the mean. In this way, it will be easier to discern the variability in CO2 concentrations, as well as the significance of the model improvements.

12. Figure 7. I cannot see the line for the biogenic sources, though it is called out in the text (Line 490).

References

G. A. Bishop, J. A. Morris, D. H. Stedman, Snowmobile contributions to mobile source emissions in Yellowstone National Park. Environ Sci Technol 35, 2874-2881 (2001); (Doi 10.1021/Es010513l).

C. M. Farkas, M. D. Moeller, F. A. Felder, B. H. Henderson, A. G. Carlton, High Electricity Demand in the Northeast US: PJM Reliability Network and Peaking Unit Impacts on Air Quality. Environ Sci Technol 50, 8375-8384 (2016); (10.1021/acs.est.6b01697).

T. D. Gordon, D. S. Tkacik, A. A. Presto, M. Zhang, S. H. Jathar, N. T. Nguyen, J. Massetti, T. Truong, P. Cicero-Fernandez, C. Maddox, P. Rieger, S. Chattopadhyay, H. Maldonado, M. M. Maricq, A. L. Robinson, Primary gas- and particle-phase emissions and secondary organic aerosol production from gasoline and diesel off-road engines. Environ Sci Technol 47, 14137-14146 (2013); (10.1021/Es403556e).

B. C. McDonald, D. R. Gentner, A. H. Goldstein, R. A. Harley, Long-term trends in motor vehicle emissions in U.S. urban areas. Environ Sci Technol 47, 10022-10031 (2013); (10.1021/es401034z).

J. Volckens, J. Braddock, R. F. Snow, W. Crews, Emissions profile from new and in-use handheld, 2-stroke engines. Atmos Environ 41, 640-649 (2007); (10.1016/j.atmosenv.2006.08.033).
* * *

---

## Referee Comment (RC2) · Anonymous Referee #2 · 20 Nov 2017

In this study, CO2 emissions for most sectors within Ontario were estimated using CO as a starting point, along with sector-specific CO2 to CO ratios from a province-wide emission inventory. Various CO2 emission inventories, including existing estimates as well as the newly-developed estimates in this study, were then used as input to a weather and chemistry/transport model to predicted CO2 concentrations. Modeled concentrations were compared against observations at four monitoring sites in Southern Ontario. Sector-specific tracking of CO2 emissions led to the conclusions that, during winter months, the daytime increase in CO2 above background was dominated by vehicle emissions, whereas at night, the increase was dominated by wintertime natural gas combustion for space heating in residential and commercial buildings.

[Figure]

The CO2 emission factor from natural gas combustion. I calculate by carbon balance that the value for pure methane should be 42 mol/m3 x 16 g/mol x 44 g CO2 / 12 g C = 2464 g CO2 / m3 of natural gas burned (assuming gas temperature of 15 deg C = 60 deg F). Real natural gas may include inert gases such as nitrogen and carbon dioxide. There may also be some incomplete combustion in the residential sector, though I expect those adjustments to be minor for the space heating sector in question. The cited emission factor (1879 g CO2/m3) may be too low, the authors should explain what assumptions underpin this emission factor, which dominates CO2 enhancements at night in their modeling.

The authors should say more about seasonality and diurnal patterns of emissions. Presumably many residential users turn down the heat at night, and some furnaces (esp. residential) run only during winter months. Have such effects been accounted for in formulating the CO2 emission inventory for southern Ontario?

It would be helpful to say more about how motor vehicle CO emissions were estimated, in particular the spatial and diurnal distribution of traffic, and also the gasoline/diesel traffic split. The use of a single CO2/CO ratio is problematic for multiple reasons. (1) the mix of gasoline versus diesel-powered vehicles varies spatially (e.g., on highway/city streets and in urban/rural areas). The diesel truck fraction tends to be much higher on major highways traveling through more sparsely populated rural areas (e.g., highway 401 outside of Toronto). The diesel CO2/CO ratio differs from the corresponding ratio for gasoline engines. Also (2) the emissions of CO are elevated during cold engine starting (and especially so during winter). Therefore the CO2/CO emissions ratio varies spatially and by time of day. The ratio should be higher on highways and lower in residential areas in the morning when vehicle engines are started under cold conditions. The method used in this study for estimating CO2 emissions from vehicles (by ratio to CO) is therefore questionable and only provides a rough approximation to a more complex reality.

Editorial suggestions:

Line 133, observational program Egbert: the word 'site' is missing

Watch sig figs in reporting emissions and calculating CO2/CO emission ratios. It is not reasonable to report emissions or emission ratios with 4-5 figures of accuracy.

Line 224: CO2 emissions should be rounded to 23.5 Mt and CO emissions should be rounded to 219 kt (even that is optimistic precision) and the ratio should be reported as 107 kt CO2/kt CO.

The same excessive precision issue is again of concern at lines 239, 247, 274-75, 283, 311, 314, 318, and in Table 2

The paper uses too many acronyms, which makes the paper harder to read. Suggest omitting some of the more obscure ones such as PIA, BBTCA, and NEE (the last one is defined on line 290 but not used anywhere else in the manuscript).

Line 359: diel shoud be diurnal

Line 365: what does TAO stand for? Since the site was operational during the period of interest for the modeling, this site should be described as part of section 2.2 rather than suddenly appearing in the manuscript at this point.

In Figure 3, the resolution is coarse and it is not easy to discern differences among the three panels shown in this Figure. The first two panels (a) and (b) are almost indistinguishable. A legend showing the color scale is missing in this Figure.

In Figures 2 and 7, the marine contribution is negligible and should be omitted to simplify these figures. The point source panel in Figure 2 is not particularly helpful either.
* * *

---

## Author Comment (AC1) · 31 Dec 2017

Interactive comments on "High-resolution quantification of atmospheric CO2 mixing ratios in the Greater Toronto Area" by S.C. Pugliese et al.

Stephanie C. Pugliese, Jennifer G. Murphy, Felix R. Vogel, Michael D. Moran, Junhua Zhang, Qiong Zheng, Craig A. Stroud, Shuzhan Ren, Douglas Worthy, Gregoire Broquet

Response to Referee #1 We thank the reviewer for their consideration of our manuscript. Our responses to their comments are given below (their original com-

[Figure]

ments are shown in small indented text).

General Comments

1. As I understand, there is provincial-level estimates of CO and CO2 that are reported by the Canadian National Inventory Report (NIR), with broad sectoral information (e.g., area, point, on-road, off-road). Then there is also a gridded inventory of CO emissions that is processed at 2.5 km x 2.5 km. The authors' reconcile the provincial-level CO2/CO estimates with the gridded CO inventory to arrive at a gridded CO2 inventory. My main critique with this approach is with the CO2/CO emission ratios reported. Some of these values seem unbelievable, which reflect potential problems in either CO2 or CO emissions reported by the NIR (more likely from CO). For example, the on-road CO2/CO emissions ratio reported here (Line 247) is 29.5 g CO2/g CO. This is equivalent to a CO emission factor of 33.9 g CO/kg CO2 or 110 g CO/kg fuel (using a carbon fraction of 0.85 g C/g fuel for gasoline). Roadway studies report tailpipe CO emission factors from gasoline cars at 10-20 g/kg fuel [McDonald et al., 2013]. The factors reported here appear too high. Also, based on the point source emission factor of 313.1 g CO2/g CO (Line 239), _0.5% of the carbon emitted is as CO and the rest from CO2. This is a very small number in the denominator from which to scale to CO2 emissions, introducing potentially large uncertainties in industrial CO2 emissions. Ultimately, I found the reporting of CO2/CO emission ratios distracting and not central to the inventory constructed. What I believe the authors' are really doing here is using the gridded CO inventory as spatial/temporal proxies for CO2 emissions, and downscaling CO2 emissions from the provincial-level to grid cells by sector. Rather than report CO2/CO emission ratios that are dubious, I suggest reframing inventory methods to emphasize the use of CO as spatial/temporal proxies for CO2.

We agree with the reviewer that the use of the CO inventory was to act as a spatial and temporal proxy for CO2 emissions. We have added text to state this to the manuscript as well as a statement that the use of CO2:CO emission ratios helps to produce realistic estimates of CO2 emissions, despite uncertainties in CO emission estimates in

lines 230-234. However, the reviewer's concern regarding the relevance of the detailed discussion of the sectoral $CO_2$:CO emission ratios also pointed out a lack of clarity in the manuscript concerning a key characteristic of the gridded CO inventory, which is that it contained separate emissions of CO for the seven primary source sectors discussed in Sections 2.4.1 to 2.4.6. Thus, there are in effect seven different spatial and temporal proxies for $CO_2$ emissions and the sectoral $CO_2$:CO emission ratios are used to weight these seven CO emissions fields. This is the explanation for the very different spatial distributions of sectoral $CO_2$ emissions evident in Figure 2. New text has been added to clarify this key aspect of the methodology in lines 207-213.

2. In Section 2.4, too many significant figures are reported in emission inventory estimates, which suggest a high degree of certainty in emissions that is unwarranted (especially for CO). Suggest using 2-3 significant figures at most.

The reviewer has raised a valid point, and we have reduced the number of significant figures used in Section 2.4 to 2-3 figures.

Specific Comments

3. Line 207. How are $CO_2$ emissions from the Canadian National Inventory Report (NIR) estimated? Are these based on energy or fuel use statistics, or from engineering calculations? Since $CO_2$ emissions is the focus of this paper, it would be helpful to include a few sentences on the basis for how $CO_2$ emissions are estimated in national reporting.

We agree with the comment and have added a description of how $CO_2$ emissions are estimated in the Canadian National Inventory Report (lines 218-222).

4. Equation 1. The "total" subscript was confusing to me. I believe what the authors' mean is "sector" in the third term of the equation (e.g., total area, point source, on-road, etc.) and "sub-sector" in the first two terms.

We have changed the subscript of the third term to "Ontario sector" to indicate the

value used is the NIR sector-wise provincial total for CO2 (kt) and CO (kt).

5. Section 2.4.1. Typically when I think of area emissions they are dispersive sources that include residential, commercial, AND industrial sources. Suggest that this source category be renamed to something like "Area industrial emissions".

We think the reviewer raises an interesting point; however, because industrial emissions are included in both the Area and Point sectors in our inventory, we have left the term "industrial" out of the category name to minimize confusion.

6. Lines 232-234. The combustion efficiency is not actually that variable, it is just that CO emissions are almost negligible from point sources and hence why the CO2:CO ratio is variable due to a tiny denominator (see Comment 1). Suggest re-wording of this sentence.

We have re-worded the sentence to highlight that uncertainties associated with very small CO emissions are likely responsible for the variable CO2:CO ratios for Point emissions (lines 253-254).

7. Lines 255-256. Lawn equipment and other small two- and four-stroke gasoline engines (e.g., snow equipment) have been shown to be a significant source of CO emissions [Gordon et al, 2013; Volckens et al., 2007; Bishop et al., 2001]. Where would they show up in the APEI, or are they included here? More importantly, how are off-road gasoline engines specifically accounted for in this study, which contribute high amounts of CO, but consume small amounts of fuel? Other off-road diesel equipment would consume significantly larger amounts of fuel than off-road gasoline engines, but have much lower CO emission factors. Not properly accounting for off-road emissions of CO between gasoline and diesel engines could affect the scaling of off-road emissions of CO to CO2.

Four examples of the kinds of sources included in the "all off-road engines" subcategory in the Off-road sector of the APEI inventory have been added to the manuscript (lines

279-280).

We were not able to separate the contributions of gasoline and diesel engines because emissions from these different sources had been aggregated during SMOKE processing of the gridded CO inventory. We agree with the reviewer that this is a challenge and limitation to our inventory and we have added a statement in the manuscript that because of this, the $CO_2$ emissions from off-road sources are an approximation of a more complex situation (lines 296-299).

8. Lines 363-365. Some more description of the FLEXPART model is needed here, or could warrant a short paragraph in the methods section. Specifically, how many hours was the back trajectory simulated for? Was this run for each site? I'm guessing the emission inventories are then multiplied by the footprint to arrive at concentrations, and then compared with ambient $CO_2$ measurements. Also, a reference to FLEXPART and some description of the model would be helpful to a reader unfamiliar with the model.

We agree more information is required about our use of the FLEXPART model. We have now outlined that footprints were generated for every third hour of the day (i.e., 00h, 03h, 06h, 09h, etc.) for the year 2014 for two sites, Downsview and TAO, and we have explained how the mixing ratio enhancements were calculated (lines 401-407). A reference has been provided to give a reader unfamiliar with FLEXPART a description of the model (line 402).

9. Line 365. This appears to be the first mention of the TAO site. It would be helpful to describe this location in more detail in Section 2.2. Also, it is not clear why looking at gradients between Downsview and TAO is a useful metric. Is it because TAO is a downwind site, whereas Downsview is downtown? In general, for a reader unfamiliar with Toronto, it would be helpful to describe locations as urban or rural more explicitly throughout Results and Discussion.

A description of TAO earlier in the manuscript in Section 2.2 was included (lines 124-126). We have also included a description why we looked at the gradient between

Downsview and TAO (as an indication of $CO_2$ mixing ratios in the downtown core of the city, since Downsview and TAO are located just north and south of the city respectively) (lines 405-407).

10. Lines 492-494. Focusing on wintertime months, while easier from a modeling perspective, would bias our understanding of $CO_2$ emissions towards wintertime sources. The sources and spatial patterns of emissions vary between winter and summer. For example, peaking plants could be important in summertime [Farkas et al., 2016]. Seems like we should try to understand both periods.

We agree with the referee that in some places there is a large variability in emissions between summer and winter months, so that peaking plants may be operational in the summer. However, 90% of the electricity generated in Ontario comes from nuclear, hydroelectric, or renewable sources so fossil-fuel peaking plants play a negligible role (http://www.energy.gov.on.ca/en/ontarios-electricity-system/ontarios-electricity-system-faqs/). Therefore, since such seasonal variability is not present in this jurisdiction, and since we are interested in understanding anthropogenic sources of $CO_2$, we have focused our study on the winter months so as to minimize the influence of the biosphere.

11. Figure 5. This plot could benefit from some uncertainty bands on the $CO_2$ measurements, such as the standard deviation or 95% confidence interval of the mean. In this way, it will be easier to discern the variability in $CO_2$ concentrations, as well as the significance of the model improvements.

We have added error bars representing the standard error of the mean to Figure 5.

12. Figure 7. I cannot see the line for the biogenic sources, though it is called out in the text (Line 490).

The Biogenic sources line is located on the zero line, underneath the Marine line. This is now explained in the text (lines 533-534).

Response to Referee #2

We thank the reviewer for their consideration of our manuscript. Our responses to their comments are given below (their original comments are shown in small indented text).

General Comments

1. The CO2 emission factor from natural gas combustion. I calculate by carbon balance that the value for pure methane should be 42 mol/m3 x 16 g/mol x 44 g CO2 / 12 g C = 2464 g CO2 / m3 of natural gas burned (assuming gas temperature of 15 deg C = 60 deg F). Real natural gas may include inert gases such as nitrogen and carbon dioxide. There may also be some incomplete combustion in the residential sector, though I expect those adjustments to be minor for the space heating sector in question. The cited emission factor (1879 g CO2/m3) may be too low, the authors should explain what assumptions underpin this emission factor, which dominates CO2 enhancements at night in their modeling.

The emission factor used in our study (1897 gCO2/m3) was estimated by the 2010 Canadian National Inventory Report, specific for the province of Ontario and based on data from a chemical analysis of representative natural gas samples and an assumed fuel combustion efficiency of 99.5 %. This information is now included in the manuscript to better explain the origin of the emission factor (lines 315-318).

2. The authors should say more about seasonality and diurnal patterns of emissions. Presumably many residential users turn down the heat at night, and some furnaces (esp. residential) run only during winter months. Have such effects been accounted for in formulating the CO2 emission inventory for southern Ontario?

In our study, the diurnal pattern of emissions was considered (Figure S2 shows the diurnal pattern of total CO2 emissions estimated by the SOCE inventory). The temporal variability of natural gas furnaces or vehicles, for example, was including in the SMOKE emissions processing system (as outlined in lines 202-207 and 320-322).

3. It would be helpful to say more about how motor vehicle CO emissions were estimated, in particular the spatial and diurnal distribution of traffic, and also the gasoline/diesel traffic split. The use of a single CO2/CO ratio is problematic for multiple reasons. (1) the mix of gasoline versus diesel-powered vehicles varies spatially (e.g., on highway/ city streets and in urban/rural areas). The diesel truck fraction tends to be much higher on major highways traveling through more sparsely populated rural areas (e.g., highway 401 outside of Toronto). The diesel CO2/CO ratio differs from the corresponding ratio for gasoline engines. Also (2) the emissions of CO are elevated during cold engine starting (and especially so during winter). Therefore the CO2/CO emissions ratio varies spatially and by time of day. The ratio should be higher on highways and lower in residential areas in the morning when vehicle engines are started under cold conditions. The method used in this study for estimating CO2 emissions from vehicles (by ratio to CO) is therefore questionable and only provides a rough approximation to a more complex reality.

We agree with the author that the use of a single CO2:CO ratio is a limitation given (1) the spatial variability of diesel-powered and gasoline-powered vehicles and (2) the temporal variability of CO2:CO for different driving-cycle phases such as cold starts in the morning. However, given that we used an existing and largely aggregated gridded CO inventory as a proxy for the spatial and temporal allocation of CO2 emissions, and used the CO inventory as the basis for estimating the CO2 inventory, we were unable to apply different emission ratios to different grid cells based on the presence of a highway or rural area nor were we able to apply a specific gasoline or diesel ratio to specific grid cells due to the lack of information on vehicle type in each grid cell. We have included some new text to the manuscript to state that based on these challenges, our estimate of On-road CO2 mixing ratios is an approximation (lines 272-276).

Editorial Suggestions

4. Line 133, observational program Egbert: the word 'site' is missing

Fixed (line 136).

5. Watch sig figs in reporting emissions and calculating CO2/CO emission ratios. It is not reasonable to report emissions or emission ratios with 4-5 figures of accuracy.

All reporting of emissions and CO2:CO ratios were reduced to 2-3 significant figures.

6. Line 224: CO2 emissions should be rounded to 23.5 Mt and CO emissions should be rounded to 219 kt (even that is optimistic precision) and the ratio should be reported as 107 kt CO2/kt CO.

The significant figures of the CO2 and CO emissions were reduced.

7. The same excessive precision issue is again of concern at lines 239, 247, 274-75, 283, 311, 314, 318, and in Table 2

All reporting of emissions and CO2:CO ratios were reduced to 2-3 significant figures.

8. The paper uses too many acronyms, which makes the paper harder to read. Suggest omitting some of the more obscure ones such as PIA, BBTCA, and NEE (the last one is defined on line 290 but not used anywhere else in the manuscript).

The acronyms PIA, BBTCA and NEE were removed from the manuscript.

9. Line 359: diel shoud be diurnal

Fixed (line 393).

10. Line 365: what does TAO stand for? Since the site was operational during the period of interest for the modeling, this site should be described as part of section 2.2 rather than suddenly appearing in the manuscript at this point.

TAO is now defined and included in Section 2.2 (lines 124-126).

11. In Figure 3, the resolution is coarse and it is not easy to discern differences among the three panels shown in this Figure. The first two panels (a) and (b) are almost indistinguishable. A legend showing the color scale is missing in this Figure.

Figure 3 was included to highlight the similarities of the FFDAS v2 inventory and the EDGAR inventory at the coarse 0.1o x 0.1o resolution, and to compare those inventories to the SOCE inventory scaled up to the same coarse resolution. The colour scale has been enlarged and moved to the bottom of the figure for easier readability.

12. In Figures 2 and 7, the marine contribution is negligible and should be omitted to simplify these figures. The point source panel in Figure 2 is not particularly helpful either.

The contributions of the Marine sector in Figures 2 and 7 are included to show its negligible contribution to $CO_2$ emissions in southern Ontario, in contrast to other areas where the influence of Marine emissions might be more significant. Consideration of this sector is also important given that two of the $CO_2$ measurement stations considered in the paper are in near-shore locations, and text noting this has been added to Section 2.4.5 (lines 306-308). Although the Point source panel in Figure 2 is not particularly helpful, emissions from this sector are significant and therefore it was included in the figure. Additionally, the Point source panel highlights the high emissions from Point sources on the western end of Lake Ontario, where the city of Hamilton, the main center for steel production in Canada, is located.

Please also note the supplement to this comment:
https://www.atmos-chem-phys-discuss.net/acp-2017-678/acp-2017-678-AC1-supplement.pdf
* * *